# Validation of a Phosphorus Food Frequency Questionnaire in Patients with Kidney Failure Undertaking Dialysis

**DOI:** 10.3390/nu15071711

**Published:** 2023-03-31

**Authors:** Joanne Beer, Kelly Lambert, Wai Lim, Ellen Bettridge, Fiona Woodward, Neil Boudville

**Affiliations:** 1Nutrition and Dietetics Department, Sir Charles Gairdner Hospital, Nedlands, WA 6009, Australia; 2School of Medical, Indigenous and Health Science, Faculty of Science, Medicine and Health, University of Wollongong, Wollongong, NSW 2522, Australia; klambert@uow.edu.au; 3Department of Renal Medicine, Sir Charles Gairdner Hospital, Nedlands, WA 6009, Australia; wai.lim@health.wa.gov.au (W.L.);; 4Nutrition and Dietetics Department, Fiona Stanley Hospital, Murdoch, WA 6150, Australia; ellen.bettridge@health.wa.gov.au; 5Nutrition and Dietetics Department, St John of God, Bunbury, WA 6223, Australia; fiona.woodward@sjog.org; 6Medical School, University of Western Australia, Nedlands, WA 6009, Australia

**Keywords:** dietary phosphorus, food frequency questionnaire, agreement, kidney failure, dialysis, validity

## Abstract

Nutritional guidelines recommended limiting dietary phosphorus as part of phosphorus management in patients with kidney failure. Currently, there is no validated phosphorus food frequency questionnaire (P-FFQ) to easily capture this nutrient intake. An FFQ of this type would facilitate efficient screening of dietary sources of phosphorus and assist in developing a patient-centered treatment plan. The objectives of this study were to develop and validate a P-FFQ by comparing it with the 24 hr multi-pass recall. Fifty participants (66% male, age 70 ± 13.3 years) with kidney failure undertaking dialysis were recruited from hospital nephrology outpatient departments. All participants completed the P-FFQ and 24 hr multi-pass recalls with assistance from a renal dietitian and then analysed using nutrient analysis software. Bland–Altman analyses were used to determine the agreement between P-FFQ and mean phosphorus intake from three 24 hr multi-pass recalls. Mean phosphorous intake was 1262 ± 400 mg as determined by the 24 hr multi pass recalls and 1220 ± 348 mg as determined by the P-FFQ. There was a moderate correlation between the P-FFQ and 24 hr multi pass recall (r = 0.62, *p* = 0.37) with a mean difference of 42 mg (95% limits of agreement: 685 mg; −601 mg, *p* = 0.373) between the two methods. The precision of the P-FFQ was 3.33%, indicating suitability as an alternative to the 24 hr multi pass recall technique. These findings indicate that the P-FFQ is a valid, accurate, and precise tool for assessing sources of dietary phosphorus in people with kidney failure undertaking dialysis and could be used as a tool to help identify potentially problematic areas of dietary intake in those who may have a high serum phosphate.

## 1. Introduction

Hyperphosphatemia in patients with kidney failure is associated with increased morbidity and mortality [1]. This may be due to an increased risk of vascular calcification, coronary artery disease, heart failure, metabolic bone disease (renal osteodystrophy), and the development of secondary hyperparathyroidism [2].

Chronic kidney disease-mineral bone disorder (CKD-MBD) was first proposed in 2006 by the Kidney Disease: Improving Global Outcomes (KDIGO) working group. This progressive disorder (distinguished from renal osteodystrophy) is related to an impaired mineral regulation that affects the homeostasis among the renal, skeletal, and cardiovascular systems [3]. It is defined as “A systemic disorder of mineral and bone metabolism due to chronic kidney disease (CKD) manifested by either one or a combination of the following: abnormalities in calcium, phosphorus, parathyroid hormone (PTH) or vitamin D metabolism, and/or, abnormalities in bone turnover, mineralization, volume, linear growth, or strength, and/or vascular or other soft tissue calcification.” [4].

The clinical features of CKD-MBD are primarily due to a compromised renal excretion of minerals leading to abnormalities in circulating calcium, phosphate, PTH, and vitamin D [5].

This presents as biochemical imbalances, increased risk of fracture, bone pain and cardiovascular events leading to poor health outcomes, and diminished quality and length of life [6].

The 2017 KDIGO guideline reiterates the complexity of CKD-MBD and recommends treatment based on assessments of phosphate, calcium, and iPTH, collectively. The recommendations also state that treatment should be based on progressively elevated serum phosphate and focus on lowering levels towards the normal range [7].

Reduction in serum phosphate levels can be achieved through more efficient dialysis, a low phosphorus diet, and the initiation of phosphate-binding medications. Phosphate binders are prescribed in almost 90% of kidney failure patients, but the non-adherence to this medication is high [8]. Consequently, management of hyperphosphatemia remains a major challenge for patients with kidney failure and the healthcare professional team.

Nutritional guidelines for phosphate management have traditionally recommended limiting dietary phosphorus [9]. However, due to the high risk of malnutrition with this approach, it must be carefully balanced with adequate energy and protein intake [10].

Dietary phosphorus is found in most foods in two forms—organic and inorganic sources. Organic sources include animal and plant-based phosphorus, which have varying degrees of bioavailability (40–60% versus 20–40%, respectively). Dietary phosphorus found in plant seeds, nuts, cereals, beans, peas, and legumes is mostly in the form of phytate or phytic acid. As humans do not have the enzyme phytase the bioavailability is lower in these food sources.

There have been increased concerns about the rising content of inorganic dietary phosphorus (phosphorus-containing additives) in modern diets. Phosphorus additives are used in food processing in many ways. For instance, to enrich or fortify foods (e.g., processed cheese), to add a raising agent (e.g., self-raising flour), to improve shelf life (e.g., biscuits), or to act as a stabiliser (e.g., in ice cream and meat products). A study by Leon et al. [11] found that 44% of the best-selling foods (from a dataset of grocery sales produced by The Neilson Company) in the US included phosphorus-containing additives. A similar Australian study, which reviewed 3000 popular food and beverage items, found that nearly half contained phosphorus-based additives [12]. These additives were particularly common in bakery goods, biscuits, and frozen and pre-prepared meals. Inorganic phosphorus sources are highly absorbable (>90%) and contribute substantially to dietary phosphorus load. A randomised control trial (RCT) of dietary counselling to reduce dietary phosphorus additive intake led to a significant reduction in serum phosphate levels by 0.70 mmol/L compared to 0.14 mmol/L in patients who did not receive this education [13]. Therefore, it would seem appropriate to develop more targeted education and counselling strategies to reduce highly absorbed dietary phosphorus whilst preserving energy and protein intake.

In the recent Kidney Disease Outcomes Quality Initiative (KDOQI) guidelines [10], the statements on phosphorus recommended that in adults with CKD 3-5D, adjusting dietary phosphorous intake is recommended to maintain serum phosphate levels in the normal range (GRADE evidence 1B). In addition, for adults with CKD 1-5D or post transplantation it is considered reasonable to consider the bioavailability of phosphorous sources (e.g., animal, vegetable, or additives) when making decisions about the nature of the dietary phosphorous restriction [10].

Unfortunately, dietary management of any kind in this cohort of patients also comes with the added challenge of limited and variable access to renal dietitians [14], logistical difficulties for patients, as well as complex and often competing medical issues and multiple medical appointments. Therefore, the ability to provide a quick and easy way to assess both organic and inorganic dietary phosphorus intake would be a valuable tool.

To deliver appropriate, timely, and targeted dietary advice an accurate diet history is critical. The multiple (7-day) weighed food record is often referred to as the ‘’gold standard” [15]. However, it is rarely used as it is time-consuming for both patient and dietitian, prone to recording bias, and can be intrusive for the patient [15]. In the clinical setting, the 24 hr multi-pass recall is utilised for measuring dietary intake instead [15]; although a single 24 hr period may not accurately reflect the individual’s usual intake. The 24 hr multi-pass recall typically takes 30–45 min to complete and needs to be performed by a trained dietitian, which may be challenging in renal services that have limited dietetic support and large patient numbers [14]. A food frequency questionnaire has the advantage of being much quicker to administer and more accurately captures habitual variability in staple foods [15]. Furthermore, in the kidney failure context, a focused, phosphorus-specific FFQ (P-FFQ) can rapidly identify foods that contain high bioavailable inorganic phosphorus, enabling targeted education even in less dietetic resource-rich environments.

Currently, there is no validated P-FFQ, and providing this would add a valuable, efficient screening tool for assessing actual dietary sources of phosphorus and assist in developing a patient-centered plan for treatment. There is no gold standard for validation of an FFQ; however, personal diet records or 24 hr multi pass recalls are often used as a reference method. Ideally, collecting a number of independent 24 hr multi pass recalls is recommended to allow estimation of intake for comparison with an FFQ [16]. Thus, the aim of this study was to validate a P-FFQ by comparing it with the 24 hr multi-pass recall method on three separate days.

## 2. Materials and Methods

### 2.1. Participants

Participants for this study were invited and recruited from nephrology outpatient departments and dialysis units at four hospitals in Western Australia between July 2021 and August 2022. A total of fifty participants (target sample), aged between forty and ninety-one years (33 males, 27 females) completed the study. The patients included those on peritoneal dialysis (PD), *n* = 4, and haemodialysis (HD), *n* = 46. This included those maintained on in-center, satellite, and home haemodialysis programs.

Inclusion criteria for the study included adult patients with kidney failure aged at least 18 years who have been maintained on dialysis (peritoneal or haemodialysis (home or facility-based)) for at least 3 months, are able to provide informed consent, have no cognitive deficits, and are able to read written English. Exclusion criteria included recent hospitalisation within the last 1 month (regardless of cause), serious intercurrent illness within the last month, and/or are unable to complete the P-FFQ or unable to provide informed consent, e.g., patients with a cognitive deficit and/or language barrier.

### 2.2. Materials

The P-FFQ assesses the participant’s intake of 31 types of food and beverages known to be high in organic and inorganic phosphorus (Appendix A). The selection of foods for inclusion in the questionnaire was based on known consumption patterns and foods of high dietary phosphorus content. Firstly, food groups were included if they contributed to at least 5% of phosphorus intake of Australian adults, as determined by the National Nutrition Survey 1995 [17]. Secondly, foods identified from the literature that contain significant amounts of bioavailable phosphorus additives, such as cola, sports drinks, and convenience meals, were included [12,18,19]. The frequency of consumption of these foods was divided into seven categories: never, less than once a month, 1–2 times a month, 1–2 times a fortnight, 2–5 times a week, once a day, and 2 or more times a day (Appendix A). Prior to the study commencement, the content validity of the P-FFQ was reviewed by five renal dietitians at the participating hospitals following which it was updated, approved, and finalised.

The validated 24 hr multi-pass recall methodology was considered to be the most appropriate reference method (shown in Appendix A) as it captures detailed information about all foods and beverages consumed by the participant in the previous 24 hr [20].

### 2.3. Data Collection

Data collection was conducted via telephone, telehealth (Zoom, Microsoft Teams, Facetime), or face-to-face, if appropriate. The predominant method of the interview was via telephone as this was preferred by many patients and dietitians and avoided any issues related to COVID-19 transmission.

In the initial consultation, after receiving written or verbal consent depending on how it was conducted (face-to-face or telehealth), participants completed the initial P-FFQ and a 24 hr multi-pass recall. Both were verified by the renal dietitian in an interview-style format to substantiate the results. This confirmed the number of serves, portion sizes, and takeaways as well as brand names of regular packaged foods. At the end of the first phase of data collection, a further two appointments were made with the participant to collect the additional dietary intake assessments. Participants completed two further 24 hr multi-pass recalls with the renal dietitian making sure that all three of the assessments covered a dialysis day, a non-dialysis weekday, and a non-dialysis weekend day. At the final (third) consultation, the P-FFQ was repeated. All information was collected on paper, scanned by the site dietitian, and sent to the principal investigator for data entry, deidentification, and analysis.

Data analysis was conducted using the Australian-specific nutrient analysis program FoodWorks V.10 (Xyris Software [Highgate Hill, QLD, Australia] Pty Ltd.) to analyse specific dietary nutrients and core food groups. Phosphorus intake from both the 24 hr multi-pass recall and P-FFQ was determined using the Australian Food and Nutrition Database (AUSNUT) and Australian Reference Database (NUTTAB) food classification system [21]. Serve sizes for the P-FFQ were quantified using standard serving sizes defined by the Australian Dietary Guidelines (2013) [22]. For instance, a serving was 1 slice of bread, ½ cup cooked porridge, ¾ cup cereal flakes, 100 g cooked protein, 2 eggs, and 250 mL milk. Using these recommended serving sizes, a P-FFQ template was designed in the nutrient analysis software program (FoodWorks, Xyris Software, Highgate Hill, QLD, Australia) to enable efficient data input and analysis. The process of data collection and analysis is shown in the study plan schematic (Figure 1).

Ethical approval for this study was obtained through Sir Charles Gairdner and Osborne Park Hospital Health Care Group with respective site-specific ethic approval from St John of God Health Care. The study was registered with the Australian and New Zealand Clinical Trial Register (ACTRN:12621000681853p). All participants provided written or verbal informed consent prior to enrolment in the study; the latter was approved as a mode of consent given the COVID-19 pandemic.

### 2.4. Statistical Analysis

Baseline characteristics of the cohort are described as number (proportion), mean (standard deviation [SD]), or median (interquartile range [IQR]) where appropriate. Normality was determined using the Shapiro–Wilk test. To reduce misreporting (implausible reporting), individuals with daily energy intake of less than 500 kcal or more than 3500 kcal were excluded from the analysis [23]. Bland–Altman analyses [24] were used to determine the agreement between the average of the two P-FFQs (applied to determine that there was not a significant intraclass variability between the two assessments) and the reference method, which in this study was considered to be the average of three 24 hr multi-pass recalls. Bland–Altman plots were constructed to show the average of the difference between the two methods and limits of agreement to identify over and underestimation. A sample size calculation of *n* = 50 was determined to have an 80% power (two-tailed, alpha = 0.05) to detect a Pearson’s correlation coefficient of 0.68 in a test of association between dietary phosphorus intake averages derived from the baseline FFQ and three 24 h multi-pass calls.

Evaluation of reproducibility of the estimation of phosphorous intake was assessed using intra-class correlation (expressed as correlation coefficient with 95% confidence intervals (95% CI)). This was used to establish the consistency and absolute agreement in the amount of phosphorous intake obtained by FFQ compared to the 24 hr multi-pass recall. Consistency and absolute agreement are derived from the intraclass correlation coefficients. To calculate the accuracy of the measurements within an acceptable standard the percent error was determined using the calculation % error = ((accepted value-measured value)/accepted value) × 100%. A percentage error cut point of ±30% has been suggested as a pragmatic cut-off for the determination of whether the proposed method is a suitably precise alternative to the reference method [25].

Descriptive statistics, intra-class correlation, and Kendall’s tau-b correlation were also performed to assess and compare the dietary intake on dialysis and non-dialysis days according to the 24 hr multi-pass recalls done on these days. Data were analysed using IBM SPSS version 26.0 (Armonk, NY, USA).

## 3. Results

Fifty participants completed the study with an age range between 40 and 91 years old. The mean duration on dialysis was 41 months ± 64 months. Table 1 shows the characteristics of the study cohort.

Based on the mean of the P-FFQ and 24 hr multi-pass recall, a total of 78% (*n* = 39) of individuals exceeded the recommended daily phosphorus intake of 1000 mg [9], 14% (*n* = 7) consumed between 800 and 1000 mg, and 8% (*n* = 4) less than 800 mg/day. Of the fifty participants, two (4%) had a difference in phosphorus intake between the 24 hr multi pass recall and a P-FFQ greater than 500 mg. This was explained by a one-off-social eating occasion (higher level recorded on 24 hr multi pass recall), which fell during the data collection period and did not reflect the usual intake.

Mean dietary phosphorus intake with the 24 hr multi pass recall was 1262 ± 400 mg, compared with 1220 ± 348 mg with the P-FFQ (*p* = 0.37, Table 2). Pearson’s coefficient indicated a moderate correlation between the P-FFQ and 24 hr multi pass recall (r = 0.62, *p* = 0.001, Table 2, Figure 2). Bland–Altman analyses were used to determine the limits of agreement and the mean difference between P-FFQ and 24 hr multi pass recall methods of all nutrients (Table 3). The mean difference in phosphorus intake between the P-FFQ and 24 hr multi pass recall was 42 mg (95% limits of agreement: 685 mg; −601 mg, *p* = 0.373), indicating good agreement between the two methods (Table 3). The intraclass correlation coefficient (ICC) was 0.764 (95% CI 0.586–0.866; *p* < 0.001), indicating good reliability (Table 4). Forty-eight of fifty (96%) phosphorus values were within the 95% limits of agreement which ranged from a lower limit of −601 mg to an upper limit of 685 mg (Figure 3). The ICC between the two P-FFQs was 0.980 (95% CI 0.965–0.989; *p* < 0.001). The precision of the P-FFQ was 3.33%, indicating suitability as an alternative to the 24 hr multi pass recall technique for estimating phosphorus intake.

Mean dietary phosphorus intake on a dialysis day was 1255 ± 465 mg and 1324 ± 406 mg on a non-dialysis day (*p* = 0.217) (Table 5). Only potassium intake was significantly different between dialysis days and non-dialysis days (*p* = 0.036). There was a moderate correlation between phosphorous intake on dialysis and non-dialysis days (r = 0.62, *p* = 0.001) as well as energy (r = 0.76, *p* = 0.001) and fibre (r = 0.76, *p* = 0.001). The intraclass correlation coefficients indicated that all nutrients showed consistency and absolute agreement between dialysis and non-dialysis days. Interpretation criteria confirmed good reliability for phosphorus, energy, and fibre (0.76, 0.86, and 0.86, respectively) and moderate reliability for protein, sodium, and potassium (0.67, 0.65, and 0.51, respectively) (Table 4).

## 4. Discussion

The results of this study confirm that the P-FFQ is a valid, accurate, and precise tool for assessing sources of dietary phosphorus in people with kidney failure on dialysis. It also showed that it was a good indicator of dietary sodium (*p* = 0.176) and protein (*p* = 0.047) but not of dietary fibre, energy, and potassium (*p* = 0.001). The latter was expected as the P-FFQ was phosphorus specific so, therefore, some food items that did not contain, or had negligible amounts of, phosphorus were not included in the questionnaire. This included some discretionary foods (e.g., honey, jam, sugar, sauces, butter, oil, potato chips, sweet pastries, and lollies), plain rice, noodles, and pasta as well as fruit and vegetables.

Accurate dietary assessment is an important aspect of the management of patients with kidney failure receiving dialysis. The Block FFQ is often used in national surveys (e.g., the national health and nutrition examination survey NHANES) [26] to assess dietary intake and to assess the health and nutrition status of various populations, including the clinical setting [16]. Other FFQs have previously been used and validated in a number of studies in the dialysis population in France, Asia, and Australia [27,28,29,30]. These studies targeted energy, protein, calcium, sodium, potassium, phosphorus, and the latter polyunsaturated fats. This study, however, is the first of our knowledge to investigate the validity of a phosphate specific FFQ in dialysis patients against a 24 hr multi pass recall.

The Bland–Altman analysis indicated that the P-FFQ and 24 hr multi pass recall estimates for the group’s mean phosphorus intake were similar. This indicates that the P-FFQ is an appropriate tool to estimate average phosphorus intake and can be used as a valuable screening method for excessive phosphorus intake. It also enables the dietitian to readily identify foods containing high bioavailable phosphorus, facilitating targeted education to control serum phosphate levels. Given the simplicity of the P-FFQ, it could also be used by non-specialist dietitians to rapidly assess dietary phosphorus intake.

The P-FFQ enables easy identification of sources of high bioavailable inorganic phosphorus as well as capturing usual intakes on dialysis and non-dialysis days, which may not be covered by a 24 hr multi pass recall. The P-FFQ also accurately identifies foods that the patient may be inappropriately restricting, possibly due to dietary misunderstandings or misinformation. For instance, the inclusion of questions regarding the consumption of wholemeal bread, cereals, nuts, or legumes may enable the clinician to identify misconceptions around these important food items. These are often avoided due to the misconception that they are high in bioavailable dietary phosphorus. Recent publications have discussed the benefits of a plant-based, wholegrain diet as the future of nutrition in kidney disease [31]. However, as Byrne et al. [32] point out, the bioavailability of phosphorus in plant-based foods is highly variable, as it is affected by phytates. More importantly, the highly bioavailable phosphorus in many food additives should be addressed.

This study had some strengths. This is the first Australian study that has validated a phosphorus specific FFQ which can be used to assess dietary intake of this nutrient in a simple and efficient way. It also has the ability to identify high bioavailable sources of phosphorus, e.g., bakery items, processed foods, and drinks, to be able to target education appropriately. In this study, we were also able to analyse (via the 24 hr multi pass recalls) the differences and similarities between intake on dialysis days and non-dialysis days in those on haemodialysis. Interestingly, there was no statistical difference between dialysis and non-dialysis days in all nutrients apart from potassium. Interclass correlation indicates a moderate to good reliability in all nutrients. Finally, the fact that the study had trained renal dietitians to assist with the completion of the 24 hr multi pass recalls and P-FFQs may have reduced the variability of the results.

Limitations of this study include the use of food sources and supplies specific to Australia and New Zealand. Ideally, a weighed food record would be used as the reference standard for validation [16], but this often changes eating behaviour from the normal and is very time-consuming and difficult for patients on dialysis, making it impractical, and it would have impeded the completion of the study. Future research should explore if the use of P-FFQ translates to improved patient-level outcomes. We are currently undertaking a randomised controlled trial utilsing this P-FFQ in the management of serum phosphate in conjunction with dietary education via digital health (ACTRN:12621000746831p).

## 5. Conclusions

In summary, the P-FFQ is a valid, accurate, simple, and practical tool to estimate the usual phosphorus intake in dialysis patients. It also enables the user to readily identify foods containing high bioavailable phosphorus, facilitating targeted education to control serum phosphate levels.

## Figures and Tables

**Figure 1 nutrients-15-01711-f001:**
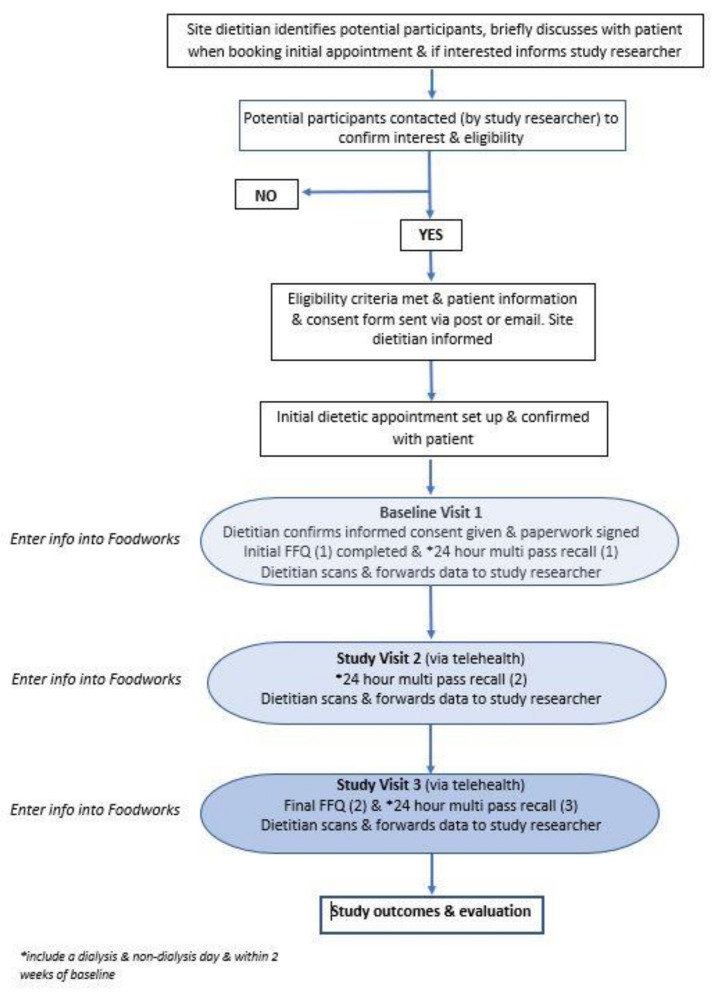
ViP kidney study plan schematic.

**Figure 2 nutrients-15-01711-f002:**
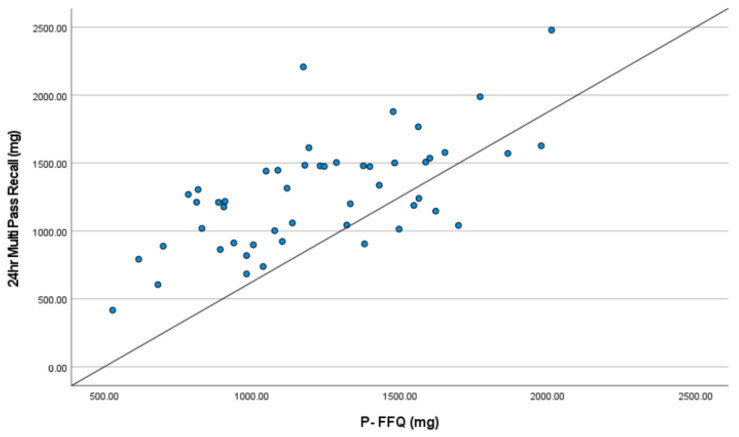
Correlation between dietary phosphorus intake according to the 24 hr multi pass recall and P-FFQ.

**Figure 3 nutrients-15-01711-f003:**
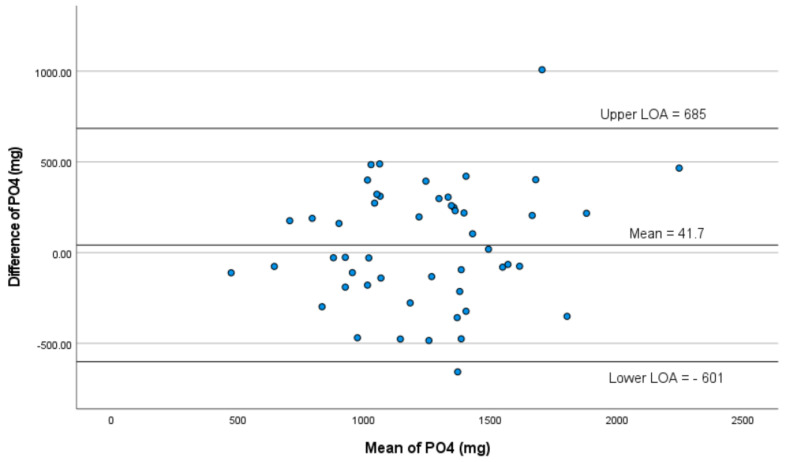
Bland–Altman plots of agreement between 24 hr multi pass recall and P-FFQ for dietary phosphorus.

**Table 1 nutrients-15-01711-t001:** Participant demographics.

Demographic	*n* = 50
Mean Age in years (standard deviation)	70 (±13.27)
Gender (*n*, %)	
Male	33 (66)
Female	17 (44)
Declared Ethnic Origin (*n*, %)	
Australian	35 (70)
New Zealand	2 (4)
European	6 (12)
Aboriginal or Torres Strait Islander	0
Asian	5 (10)
Other	2 (4)
Smoking Status (*n*, %)	
Never	35 (70)
Former	13 (26)
Current	2 (4)
ESKF Cause (*n*, %)	
Diabetic Nephropathy	14 (28)
Hypertension/Vascular	8 (16)
Polycystic Kidney Disease	5 (10)
Other *	19(38)
Unknown	4 (8)
Dialysis Type (*n*, %)	
Home Haemodialysis	2 (4)
Satellite Haemodialysis	44 (88)
Peritoneal Dialysis	4 (8)
Vintage (*n*, %)	
<12 months	20 (40)
12–24 months	10 (20)
>24 months	20 (40)
Address (*n*, %)	
Metro	42 (84)
Regional	8 (16)
Comorbidities (*n*, %)	
Diabetes	22 (44)
Peripheral Vascular Disease	8 (16)
Cardiovascular Disease	9 (18)
Coronary Artery disease	15 (30)
Cancer (NOS)	7 (14)

***** See Appendix A.

**Table 2 nutrients-15-01711-t002:** Correlations between dietary intake according to 24 hr multi pass recall and P-FFQ (*n* = 50).

	24 hr Multi-Pass Mean ± SD	P-FFQ Mean ± SD	Paired T Test *p*-Value	Pearson’s	
				*r*-Value	*p*-Value
Energy (kJ)	7403 ± 2344	5506 ± 1775	0.001	0.531	0.001
Protein (g)	82 ± 26	76 ± 22	0.047	0.554	0.001
Fibre (g)	20 ± 8	11 ± 5	0.001	0.408	0.003
Sodium (mg)	2169 ± 911	1962 ± 765	0.176	0.205	0.152
Potassium (mg)	2339 ± 684	1505 ± 489	0.001	0.563	0.001
Phosphorus (mg)	1262 ± 400	1220 ± 348	0.373	0.623	0.001

**Table 3 nutrients-15-01711-t003:** Bland–Altman Index for Limit of Agreement (LoA) and mean difference between P-FFQ and 24 hr multi pass recall (*n* = 50).

	Mean Difference	95% LoA Lower Limit; Upper Limit	Within 95% LoA
Energy (kJ)	1897	−2135; 5926	48/50 (96%)
Protein (g)	6.78	−38.7; 52.25	49/50 (98%)
Fibre (g)	9.16	−5.58; 23.9	49/50 (98%)
Sodium (mg)	206.5	−1876; 2289	48/50 (98%)
Potassium (mg)	834	−292; 1960	47/50 (94%)
Phosphorus (mg)	42	−601; 685	48/50 (96%)

**Table 4 nutrients-15-01711-t004:** Intraclass correlation coefficient and Kendall’s tau-b of dialysis and non-dialysis days according to 24 hr multi pass recall (*n* = 46, those on PD not included *n* = 4).

	Intraclass Correlation *	95% Confidence Interval	*p*-Value	Kendall’s tau-b Correlation
Energy (kJ)	0.863	0.753–0.924	0.001	0.559
Protein (g)	0.696	0.451–0.832	0.001	0.359
Fibre (g)	0.864	0.755–0.924	0.001	0.634
Sodium (mg)	0.650	0.364–0.807	0.001	0.325
Potassium (mg)	0.509	0.135–0.724	0.007	0.264
Phosphorus (mg)	0.763	0.574–0.868	0.001	0.387

* Interpretation criteria for ICC 0.50–0.75 = moderate reliability; 0.75–0.90 good reliability.

**Table 5 nutrients-15-01711-t005:** Descriptive statistics of dietary intake of dialysis and non-dialysis days according to 24 hr multi pass recall (*n* = 46, those on PD not included *n* = 4).

	Dialysis Day Mean ± SD	Non-Dialysis Day Mean ± SD	*p*-Value	*r*-Value	*p*-Value
Energy (kJ)	7387 ± 2358	7689 ± 2380	0.217	0.761	0.001
Protein (g)	83 ± 31	86 ± 27	0.475	0.535	0.001
Fibre (g)	19 ± 9	20 ± 8	0.208	0.764	0.001
Sodium (mg)	2171 ± 838	2231 ± 1036	0.673	0.448	0.001
Potassium (mg)	2191 ± 813	2478 ± 779	0.036	0.359	0.140
Phosphorus (mg)	1255 ± 446	1324 ± 406	0.217	0.622	0.001

## Data Availability

The data presented in this study are available on request from the corresponding author.

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
