# Peer review of "Validation of a Phosphorus Food Frequency Questionnaire in Patients with Kidney Failure Undertaking Dialysis"

_nutrients, 2023, doi:10.3390/nu15071711_

Round 1

Reviewer 1 Report

The validated 24-hour multi pass recall methodology is not presented in Supplemental Table 2.

Other ESKF causes presented instead

Author Response

Thank you for your feedback. Full details have been provided in the reviewers letter, but for your convenience here is a summary:

  1. The methodology has been improved to include a more thorough explanation of the study with emphasis on data collection.
  2. The 24 hour multi pass recall table has now been correctly inserted and labelled table 2 in the supplemental documents. The patient demographics - other has been correctly labelled table 3.

Kind regards,

JB

Reviewer 2 Report

Comments of this reviewer are:

- The authors should mention the limitations of the study, especially that is not validated against the “gold standard” multiple (7 day) weighed food record, in the abstract

- The Supplemental Table 2 refers to “Participant Demographics ESKF Cause – Other” and not the validated 24-hour multi pass recall methodology. Please provide the right table

- Abstract: “There was a moderate correlation between the P-FFQ and 24-hour recall (r=0.62, P=0.37)” I cannot see this in the results section.  Please clarify.

- It is not clear which P-FFQ (baseline or final) was used in the analysis.  In addition, it is not clear the reason for performing two P-FFQs.  Please clarify.

Author Response

Thank you for your review and feedback. Full details of the revisions are given in the response letter but for your convenience I have also summarised below:

The methodology and results sections have been expanded to clarify queries raised. In particular data collection, additional information and statistics in results and conclusions. The introduction has also been improved to emphasis the need for timely efficient nutritional intervention.

The following section is our replies to your more detailed questions:

Point 1: The authors should mention the limitations of the study, especially that is not validated against the “gold standard” multiple (7 day) weighed food record, in the abstract.

Response 1: This is a very important point and limitations of the study regarding the decision to use the 24 hour multi pass over the multiple 7 day weighted food record have been addressed in the introduction and the discussion. We haven’t adderd this to the abstract as this could take us over the recommended word count and is not normal practice.

Point 2: The Supplemental Table 2 refers to “Participant Demographics ESKF Cause – Other” and not the validated 24-hour multi pass recall methodology. Please provide the right table.

Response 2: The 24 hour multi pass table has now been correctly inserted and labelled as table 2. The Participant Demographics ESKF Cause – Other has now been correctly relabelled table 3.

Point 3: Abstract: “There was a moderate correlation between the P-FFQ and 24-hour recall (r=0.62, P=0.37)” I cannot see this in the results section. Please clarify.

Response 3: This has been updated and clarified in the results section.

Point 4: It is not clear which P-FFQ (baseline or final) was used in the analysis. In addition, it is not clear the reason for performing two P-FFQs. Please clarify.

Response 4: The analysis used the average of the two P-FFQ’s which in most instances was the same (45/50). The rational for doing the P- FFQ at baseline and final was to check there was not a significant intraclass variablity between the two assessments. This has been clarified in the statistical analysis and results section.

Kind regards

JB

Round 2

Reviewer 2 Report

Comments of this reviewer are:

- Table 2 has two p-values.  Why? Where do they refer? Please clarify

- “Mean dietary phosphorus intake with the 24-hour multi pass recall was 1262mg ± 400 240 mg, compared to 1220mg ± 348mg with the P-FFQ (r= 0.63, P= 0.001, Table 2, Figure 2). 241 There was a moderate correlation between the P-FFQ and 24-hour recall (r=0.62, P=0.37, 242 Table 2).” It is not clear if these two sentences refer to two different correlations.  Please clarify.

- Please check again the supplementary tables.  I still see only two supplementary tables

Author Response

Thank you again for your prompt response and highlighting some additional errors. I have corrected the manuscript and summarise the points for you here:

  1. I have clarified table 2 to show that one of the p - values relates to the paired t test and the other to Pearson's coefficient.
  2. I have also updated the sentence in the results section (page 7 line 242) to clarify the correlation between the 24hr recall and P-FFQ.
  3. I had updated the supplemental tables to include all 3 tables and have resubmitted it in case it got overlooked.

Kind regards

JB